# Quantitative CT imaging features for COVID-19 evaluation: The ability to differentiate COVID-19 from non- COVID-19 (highly suspected) pneumonia patients during the epidemic period

**Shengkun Peng[1☯], Lingai Pan[2☯], Yang Guo[2☯], Bo Gong[3], Xiaobo Huang[2], Siyun Liu[4], Jianxin Huang[5‡], Hong Pu[1‡]\*, Jie Zeng[6‡]**

1 Department of Radiology, Sichuan Academy of Medical Sciences & Sichuan Provincial People's Hospital, University of Electronic Science and Technology of China, Chengdu, China, 2 Department of Critical Care Medicine, Sichuan Academy of Medical Sciences & Sichuan Provincial People's Hospital, University of Electronic Science and Technology of China, Chengdu, China, 3 The Key Laboratory for Human Disease Gene Study of Sichuan Province, Sichuan Provincial People's Hospital, University of Electronic Science and Technology of China, Chengdu, China, 4 GE Healthcare (China), Beijing, China, 5 Department of Anesthesiology, Sichuan Academy of Medical Sciences & Sichuan Provincial People's Hospital, University of Electronic Science and Technology of China, Chengdu, China, 6 Department of Cardiology, Sichuan Academy of Medical Sciences & Sichuan Provincial People's Hospital, University of Electronic Science and Technology of China, Chengdu, China

☯ These authors contributed equally to this work.
‡ JH, HP and JZ also contributed equally to this work.
\* ph196797@163.com

**Data Availability Statement:** Data of COVID-19 patients contain potentially identifying or sensitive

## Abstract

### Objectives

COVID-19 and Non-Covid-19 (NC) Pneumonia encountered high CT imaging overlaps during pandemic. The study aims to evaluate the effectiveness of image-based quantitative CT features in discriminating COVID-19 from NC Pneumonia.

### Materials and methods

145 patients with highly suspected COVID-19 were retrospectively enrolled from four centers in Sichuan Province during January 23 to March 23, 2020. 88 cases were confirmed as COVID-19, and 57 patients were NC. The dataset was randomly divided by 3:2 into training and testing sets. The quantitative CT radiomics features were extracted and screened sequentially by correlation analysis, Mann-Whitney $U$ test, the least absolute shrinkage and selection operator (LASSO) logistic regression (LR) and backward stepwise LR with minimum AIC methods. The selected features were used to construct the LR model for differentiating COVID-19 from NC. Meanwhile, the differentiation performance of traditional quantitative CT features such as lesion volume ratio, ground glass opacity (GGO) or consolidation volume ratio were also considered and compared with Radiomics-

patient information. Data are available from the Sichuan Academy of Medical Sciences & Sichuan Provincial People's Hospital Institutional Data Access / Ethics Committee which can be contacted via scsrmyyyb@163.com and drhuangxb@163. com.

**Funding:** This research was supported by a grant from Sichuan Province Science and Technology Support Program to BG (Grant number: 2020YFS0558). GE Healthcare provided support in the form of salaries for author SL. The specific roles of the authors are articulated in the 'author contributions' section. The funders had no role in study design, data collection and analysis, decision to publish, or preparation of the manuscript.

**Competing interests:** The authors have read the journal's policy and the authors of this manuscript have the following competing interests: SL is a paid employee of GE HEalthcare. There are no patents, products in development or marketed products associated with this research to declare. This does not alter our adherence to PLOS ONE policies on sharing data and materials.

**Abbreviations:** AUC, Area Under Curve; CAP, Community-acquired pneumonia; CT, Computed tomography; GGO, Ground glass opacity; GLCM, Gray Level Co-occurrence Matrix; GLDM, Gray Level Dependence Matrix; GLRLM, Gray Level Run Length Matrix; GLSZM, Gray Level Size Zone Matrix; MDT, Multidisciplinary collaboration team; NC, Non-Covid-19; NGTDM, Neighboring Gray Tone Difference Matrix; NSIP, Non-specific interstitial pneumonia; ROC, Receiver operating characteristic; UIP, Common interstitial pneumonia.

based method. The receiver operating characteristic curve (ROC) analysis were conducted to evaluate the predicting performance.

## Results

Compared with traditional CT quantitative features, radiomics features performed best with the highest Area Under Curve (AUC), sensitivity, specificity and accuracy in the training (0.994, 0.942, 1.0 and 0.965) and testing sets (0.977, 0.944, 0.870, 0.915) (Delong test, $P < 0.001$). Among CT volume-ratio based models using lesion or GGO component ratio, the model combining CT lesion score and component ratio performed better than others, with the AUC, sensitivity, specificity and accuracy of 0.84, 0.692, 0.853, 0.756 in the training set and 0.779, 0.667, 0.826, 0.729 in the testing set. The significant difference of the most selected wavelet transformed radiomics features between COVID-19 and NC might well reflect the CT signs.

## Conclusions

The differentiation between COVID-19 and NC could be well improved by using radiomics features, compared with traditional CT quantitative values.

## Introduction

Corona Virus Disease 2019 (COVID-19) has threaten the public health. Computed tomography (CT) is one of the most intuitive assessment tools for lung disease. During pandemic, CT has played an active role in finding lesions, evaluating the severity and prognosis of disease and differentiating between COVID-19 and other pneumonia. However, the current CT diagnosis mainly depends on the experience of clinicians according to the qualitative characteristics, which may lack consistency, repeatability and operability.

The qualitative CT manifestations might be not enough for differential diagnosis among different pneumonia with imaging overlaps. Most of Non-COVID-19 (NC) patients are diagnosed as community-acquired pneumonia (CAP) during pandemic [1]. CAP is the third most common cause of death worldwide. Common pathogens of CAP are Streptococcus pneumonia, Haemophilus influenza, influenza virus, RSV virus and so on. The outbreak of COVID-19 in China was at a time of high CAP incidence. At the same period in the past, influenza viruses often spread [2]. Influenza virus and new coronavirus have similar clinical manifestations, such as cough, fever and myalgia. Laboratory tests also present a decrease in white blood cells and lymphocytes. In addition, there still lack of uniform CT imaging standards to distinguish COVID-19 from NC. At present, the most common CT manifestations of COVID-19 pneumonia are: GGO, consolidation, mixed density and subpleural distribution. However, these signs often appear in patients with bacterial infections or other viral infections. Previous report have shown that the pathological features of COVID-19 greatly resemble those seen in SARS and Middle Eastern respiratory syndrome (MERS) coronavirus infection, which makes the identification of CAP and COVID-19 particularly difficult [3]. Although, RT-PCR testing is the current gold standard for the diagnosis of COVID-19. In the current emergency, the low sensitivity of RT-PCR implies that many COVID-19 patients may not be identified and may not receive appropriate treatment in time. Previous research supports the use of chest CT for

screening for COVD-19 for patients with clinical and epidemiologic features compatible with COVID-19 infection particularly when RT-PCR testing is negative [4–6].

Under such circumstances, CT imaging is gradually required to transform from qualitative to quantitative, which is to discover the invisible difference under the veil of similar CT signs or quantify the manually interpreted CT signs. The digital medical images contain a large amount of invisible deep information. With the rapid development of computing power and algorithms, more and more useful quantitative imaging indicators could be derived and applied in clinical studies. Radiomics is a method that covers a large number of quantitative imaging features, from first-order statistical features to second or transformed higher-order features [7]. The radiomics has been widely utilized in tumors, which can indicate genotyping, tumor microenvironment and susceptibility [8]. Several texture-based studies in pneumonia have also been reported including: identification of lung tissue, differentiation between different types of pneumonia, discrimination of pneumonia and other lung diseases, and prognosis prediction of pneumonia [9–14]. However, there is still a lack of effective quantitative features or standard quantitative criteria for the differentiation between COVID-19 and NC but highly suspected patients.

Therefore, the current study aimed to evaluate the effectiveness of radiomics-based quantitative CT features in discriminating COVID-19 from highly suspected NC. Meanwhile, the differentiation performance of traditional quantitative CT features such as lesion volume ratio, GGO or consolidation volume ratio were also considered and compared with radiomics-based method, which was to comprehensively assess the value of quantitative CT imaging analysis in pneumonia diagnosis.

## Methods

### Patients

We have abided by ethical standards of journal and the study design was approved by institutional Ethics Committee of Sichuan Academy of Medical Sciences & Sichuan Provincial People's Hospital. We retrospectively enrolled 1988 fever patients from four centers of fever clinic in Sichuan Province during the period of January 23 to March 23. 1781 patients were excluded which showed negative lung imaging and COVID-19 nucleic acid test and without epidemic exposure history. Data including severe CT imaging artifacts or contrast-enhanced CT images were also excluded. Among the rest population, 207 patients had positive pulmonary manifestations, of which 88 patients showed positive lung imaging with positive COVID-19 nucleic acid test. 119 of them were suspected COVID-19 pneumonia patients, who present initially negative COVID-19 nucleic acid test, but were identified as highly suspected who cannot be exclude from COVID-19 (Fig 1) according to CT manifestations and consultation by multidisciplinary team (MDT) (specialists from infection, radiology and respiratory department, and finally excluded by repeated nucleic acid test.) While, 62 patients were excluded from 119 suspected patients, because there was no clear microbiological evidence. Finally, 57 non-COVID-19 patients were selected with microbiological evidence of infection including influenza A or B virus infection, bacterial infection and mixed-type infection. The CT images, laboratory indicators and clinical characteristics were recorded in Table 1.

All patients underwent non-enhanced chest spiral CT scanning. We selected the initial CT images scanned when the patients were first admitted to the hospital. Images from all cases were collected and analyzed by two radiologists (H. P [a thoracic radiologist with 28 years' experience] and SK. P [a radiology attending doctor with 7 years' experience in interpreting chest CT images]). The results of the judgement were reached by consensus.

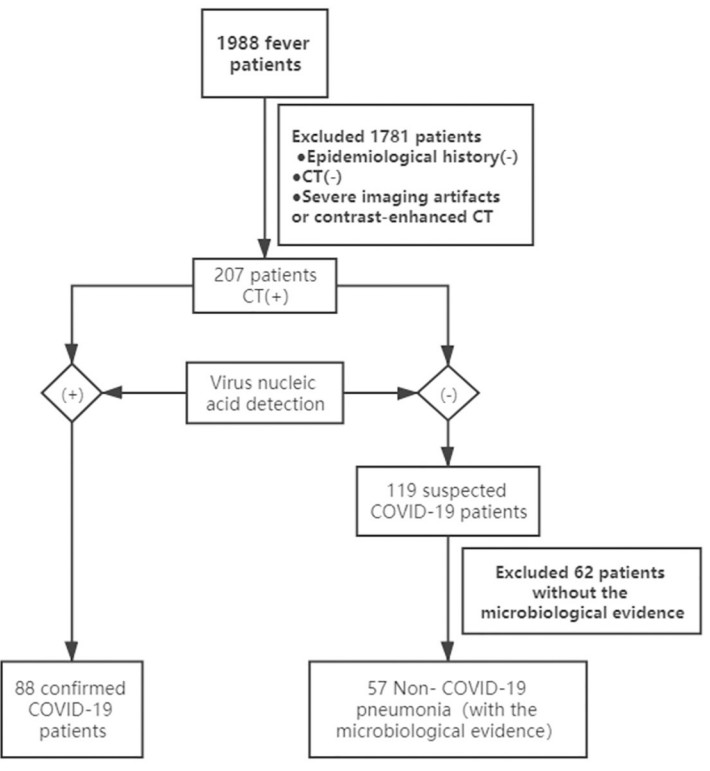

**Fig 1. Screen flow chart.**

## CT image acquisition

All patients underwent non-enhanced chest spiral CT scanning. We selected the initial CT images scanned when the patient admitted to the hospital. The CT scanners were as follows: UCT528 40-row 40-slice spiral CT (United Imaging, shanghai, china), Siemens 16-row CT (Emotion; Germany) and Siemens 64-row (SOMATOM Perspective); and the following scanning parameters were used: slice thickness 1.0mm or 2mm, tube voltage 130 kVp, tube current

**Table 1. Demographics of patients infected with COVID-19 and non-COVID-19 patients.** Data are represented as n (%) or mean (SD).

|  | COVID-19 | | NC | |
|---|---|---|---|---|
|  | Training set(n = 52) | Testing set(n = 36) | Training set (n = 34) | Testing set (n = 23) |
| Age | 44.5(14.4) | 47.9(14.5) | 42.1(16.3) | 42.2(16.4) |
| Gender/male | 25(48%) | 23(64%) | 18(53%) | 17(74%) |
| Symptoms |  |  |  |  |
| Fever | 12(23%) | 7(19%) | 26(76%) | 17(74%) |
| Cough | 13(25%) | 13(36%) | 23(68%) | 14(61%) |
| Myalgia or fatigue | 11(21%) | 7(19%) | 7(21%) | 7(30%) |
| Dizzy | 5(10%) | 2(6%) | 2(6%) | 3(13%) |
| Diarrhea | 1(2%) | 2(6%) | 1(3%) | 0 |
| Sore Throat | 1(2%) | 2(6%) | 5(15%) | 4(17%) |
| Headache | 0 | 1(3%) | 3(9%) | 1(4%) |
| Chest pain | 2(4%) | 0 | 1(3%) | 0 |
| Hospital length of stay(day) | 14.4(5.3) | 13.8(4.6) | 6.1(4.1) | 5.3(3.3) |

30mAs or tube voltage 120 kVp, tube current was regulated by an automatic exposure control system(80 to 300 mAs). The patient was examined in the supine position. The scan range was from the entrance of the chest to the lower lung (costophrenic angle), continuous screening with single breath at the end of inspiration. Images were reconstructed with a slice thickness of 1.5 mm or 1 mm and an interval of 1.5 mm or 1 mm, respectively. All images were displayed with standard lung (width, 1200 HU; level, –600 HU) and mediastinal (width, 350 HU; level, 40 HU) window settings. All images were exported as digital imaging and communications in medicine format for quantitative image feature extraction.

## Image analysis

The workflow of imaging analysis and data processing was illustrated in Fig 2. The CT image analysis was conducted by Lung Kit software (GE Healthcare, Version LK2.2) and the detailed procedure was described in the Supplementary Method in S1 File. To reduce the impact of different scanner and scanning parameters, all the images were firstly resampled into isotropic voxel size of 1 mm*1 mm*1 mm using trilinear interpolation. After interpolation into isotropic voxel, the image's intensity values were rounded to the nearest integer HU value. The low-pass Gaussian filter with σ = 0.5 was then conducted to increase the reproducibility of the radiomics features to be extracted [15]. Afterwards, the five anatomic lung lobes were firstly automatically segmented, which was followed by automatic pneumonia lesion volume of interest (VOI). The margin of the VOI was checked and manually adjusted by an experienced thoracic radiologist (SK. P, a radiology attending doctor with 7 years' experience in interpreting chest CT images) and the obviously swollen blood vessels involved in the lesion were excluded, if necessary. All the automatically segmented or manually adjusted VOIs were checked by a senior radiologist (H. P [a thoracic radiologist with 28 years' experience]) to reach consensus. The distributed lesions were considered as a whole VOI in the following analysis steps.

Following the CT value discretized with binWidth = 25HU, a total of 851 radiomics features were then extracted from segmented VOIs by using Python package Pyradiomics v2.2 (The formula for calculating each feature were described in the website: https://pyradiomics. readthedocs.io/en/latest/index.html) [16]. In brief, a total of 851 extracted radiomics features were categorized into three groups: first-order features, textural features, and transformed features. There were 32 first-order features consisting of 18 intensity statistical features and 14 morphological features. Among 75 textural features, there were 24 Gray Level Co-occurrence Matrix (GLCM), 16 Gray Level Run Length Matrix (GLRLM), 16 Gray Level Size Zone Matrix (GLSZM), 14 Gray Level Dependence Matrix (GLDM) and 5 Neighboring Gray Tone

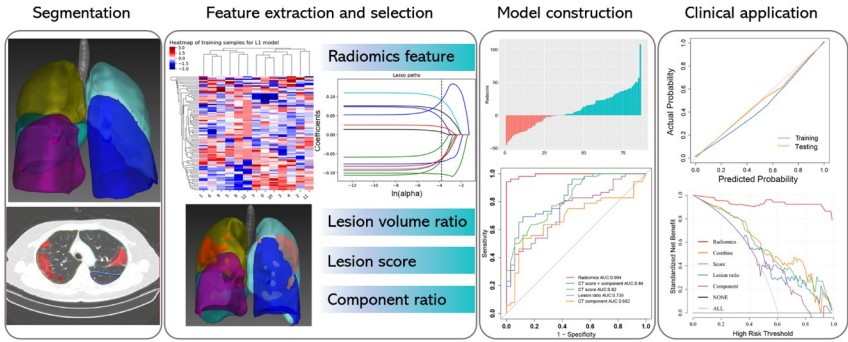

**Fig 2. Workflow of CT imaging analysis.**

Difference Matrix (NGTDM) features. For the transformed images, first-order wavelet transform decomposed the VOI into 8 sub-ROIs by using either a high- or low-pass filter in three dimensional directions, including high–high–high, high–high–low, high–low–low, high–low–high, low–high–low, low–high–high, low–low–high, and low–low–low. The same set of texture features (18 intensity statistical features, 24 GLCM features, 16 GLRLM, 16 GLSZM, 14 GLDM, and 5 NGTDM) were calculated based on these wavelet-transformed images and 744 wavelet features were obtained finally. These common or wavelet-transformed radiomics features were mainly calculated as referring to the previous studies on pneumonia differentiation [7, 8].

For lesion volume ratio analysis, the lesion volume ratio in five lung lobes was respectively calculated automatically by LK2.2 after lesion VOI was delineated. The lesion ratio in each lung lobe was scored from 0 to 5 which was defined according to the volume ratio involved: 0, no lesion; 1, ≤5%; 2, 6%-25%; 3, 26%-49%; 4, 50%-75%; 5, >75% [17].

For lesion component analysis, the part solid component was defined as the lesion with CT value <-200 HU while the parts with CT value >-200 HU were defined as solid component. And the volume ratio of each kind of lesion component in the left and right lung was derived automatically.

CT Image processing results and manually interpreted CT signs were aligned by two radiologists (H. P [a thoracic radiologist with 28 years' experience] and SK. P [a radiology attending doctor with 7 years' experience in interpreting chest CT images]). The results of the judgement were reached by consensus.

## Data preprocessing, model establishment and evaluation

The radiomics feature data was firstly preprocessed by replacing missing values with median values and z-score normalization. The dataset (n = 145; COVID-19 = 88, NC = 57) was randomly stratified into training (n = 86) and testing (n = 59) set using the ratio of 3:2.

Following the CT image analysis and data preprocessing, five logistic regression models were constructed for differentiating COVID-19 from NC. Model A was constructed by radiomics features; Model B was constructed by lesion volume ratio in five lung lobes; Model C was constructed by lesion ratio score in each lung lobe derived from lesion volume ratio; Model D was constructed by volume ratio of lesion part-solid and solid component in left and right lungs; Model E was combined model using lesion score and component ratio. For every model, the features were selected sequentially only in the training dataset because the distribution of test dataset was taken as unknown status.

For Model A, the radiomics features for model construction were selected by the procedure as follows. The redundant features were firstly reduced by correlation analysis at a cut-value of 0.6. Then Mann-Whitney *U* test was used to eliminate features without statistical difference in COVID-19 and NC groups. The significant level was 0.05. Next, the least absolute shrinkage and selection operator (LASSO) logistic regression method with 10-fold cross validation was used for further feature selection and regularization to improve the model accuracy and avoid overfitting. The remaining features with non-zero coefficients were kept and involved into multi-variate backward stepwise logistic regression with minimum AIC (Akaike Information Criterion) method to establish the model. Besides, to verify the reliability of selected radiomics features in the logistic regression model, 100-times bootstrapping and 100-fold leave-group-out cross-validation (LGOCV) were performed.

From Model B to Model D, the features of lesion ratios or scores were sequentially selected by Mann-Whitney *U* test and univariate logistic analysis. The features with statistically P<0.05 were used for respective logistic regression model construction and the model score was

derived from Eq (1). And for combined Model E, the model score of Model C and Model E derived from logistic regression were involved directly to construct logistic regression model.

$$model\ score = \beta_0 + \beta_1 x_1 + \beta_2 x_2 + \cdots + \beta_n x_n \tag{1}$$

in which $\beta_0$ is the constant, $\beta_i$ is logistic regression coefficient and $x_i$ is the value of selected features in the model.

The five logistic regression models were finally constructed based on the training set. And models' classification performances were evaluated by receiver operating characteristic (ROC) analysis. The area under the ROC curve (AUC), accuracy, sensitivity and specificity were derived. To internally validate the model performances in the testing set, the retained feature names, logistic regression coefficients and the ROC cut-off value (Youden index reached the maximum) obtained from the training set were applied in the testing set. Then the corresponding performance in the test data could be calculated. In addition, the calibration curves and decision curve analysis (DCA) curves were calculated to assess the models' prediction performance and clinical benefit.

### Statistical analysis

The continuous variables or ordinal variables were compared by t-test or Mann-Whitney $U$ test. The distribution of different CT manifestations judged by radiologist in COVID-19 and NC groups were compared by chi-squared test or Fisher exact test when small sample sizes occurred. When three or more manifestation types were included and compared by chi-squared test, the adjusted standardized residuals was calculated to characterize cell significance and the absolute value of adjusted residual larger than 2.58 was considered to be significant [18]. The Cramer's V coefficients [0.1 (small); 0.3 (medium); 0.5 (large)] were used to assess the association between the categorical variables [19]. For ROC analysis, the cut-off value in the training set at the maximum of Youden index of each model was calculated and the sensitivity, specificity, accuracy in the training and testing sets were derived at such cut-off value. The ROC curves between different models were compared by Delong test. The reported statistical significance levels were all two-sided, and the statistical significance was set at 0.05. All these statistical analyses were conducted with SPSS Software (Version 25, IBM, Chicago, IL), GraphPad Software (version 8, San Diego, CA) and R software (Version: 3.6.1, https://www.r-project.org). The following R packages were mainly utilized: "glmnet" for logistic regression including LASSO algorithm; "pROC" for ROC analyses; "rmda" for DCA analysis.

## Results

### Feature selection

**1. Radiomics feature selection.** Among 851 radiomics features, by using Spearman's correlation coefficient cut-value of 0.6, 47 features with low correlation were remained. Then, 26 features which were significantly different between COVID-19 and NC groups (P < 0.05) were retained after Mann-Whitney $U$ test. Afterwards, through univariate logistic regression analysis (18 features left) which was followed by 10-fold cross validation of LASSO logistic regression algorithm, 13 features with non-zero coefficients were retained and involved into multi-variate backward stepwise logistic regression. With minimum AIC (Akaike Information Criterion) method, 10 features were finally selected to establish the radiomics Model A, which included wavelet.LHL_firstorder_Skewness (wLHL_fS), wavelet. HHL_glcm_Idn (wHHL_glcm_Idn), wavelet.LLL_glszm_SizeZoneNonUniformityNormalized(wLLL_glszm_SZNUN), wavelet.LLH_glcm_InverseVariance (wLLH_IV), wavelet.

HLH_gldm_DependenceNonUniformityNormalized (wHLH_gldm_DNUN), wavelet.
HLH_glszm_SmallAreaEmphasis(wHLH_glszm_SAE), original_shape_Flatness (ori_Flat-
ness), original_glszm_SizeZoneNonUniformity (ori_glszm_SZNU), original_firstorder_10-
Percentile(ori_f10P), wavelet.HHH_glcm_Imc1 (wHHH_glcm_Imc1). The LASSO feature
selection procedure was shown in S1(A) and S1(B) Fig in S1 File.

The statistical differences of each radiomics feature between COVID-19 and NC groups in
training and testing sets were illustrated in S2 Fig in S1 File and summarized in S1 Table in S1
File. The wavelet-HHL_glcm_Idn, wavelet-LLL_glszm_SizeZoneNonUniformityNormalized,
wavelet-LLH_glcm_InverseVariance and, wavelet-HHH_glcm_Imc1features showed signifi-
cant differences (P<0.05) between COVID-19 and NC in both of training and testing sets.

**2. Lesion ratio feature selection.** The lesion volume ratio, score and component features
were also selected to construct Model B to Model D. The lesion volume ratio of left and right
bottom lung lobes were selected as significant features in Model B. The lesion-volume-based
score in five lung lobes were all included in the Model C. And for Model D using component
volume ratio, the GGO volume ratios in left and right lungs were selected. The selected fea-
tures in each model and their statistical differences between COVID-19 and NC in training
and testing sets were summarized in S2 Table in S1 File. It showed that all these lesion ratio
features were significantly different (P<0.05) in COVID-19 and NC groups. The bar plot of
lesion volume ratio and component volume ratio in different lung lobes were illustrated in S3
and S4 Figs in S1 File.

## Model establishment and evaluation

The logistic regression calculation formula constructing Model A to Model E were shown in
Table 2. The classification score of each model derived by weighting logistic regression coeffi-
cients onto selected features were illustrated for COVID-19 and NC, as shown in S5 Fig in S1
File. The confusion matrix and the classification score of each model were illustrated in Fig 3.

By ROC analyses, the performances of Model A to Model E were summarized in Table 3
and the ROC curves and precision-recall curves were respectively illustrated in S6 and S7 Figs
in S1 File. Among five differentiation models, the Model A using radiomics features performed
best with the highest AUC, sensitivity, specificity and accuracy in the training and testing sets.
The AUC, sensitivity, specificity and accuracy were 0.994 (0.984–1.0), 0.942 (0.8375–0.9862),
1.0 (0.8793–1.0) and 0.965 (0.8982–0.9923) in the training set, while the corresponding values
in the testing set were 0.977(0.947–1), 0.944 (0.8091–0.9941), 0.870 (0.6703–0.9631), 0.915
(0.8125–0.9673), respectively.

In addition, as shown in S7 Fig in S1 File, all the model performance indicated by preci-
sion-recall curves shared the same tendency as the ROC analysis. The radiomics model (Model
A) performed better than other image-derived models both in the training and test group,
which was followed by combined model (Model E). It indicated the robustness of the radio-
mics model. Moreover, 100-times bootstrapping and 100-fold leave-group-out cross-valida-
tion (LGOCV) were also used to verify the reliability of the selected radiomics features and the
regression model (detail in Supplementary Method in S1 File). Except for wavelet. LHL_first-
order_Skewness, all other features appeared more than 50 times during multiple splits into
training and testing set as shown in S3 and S4 Tables in S1 File. The estimation of model over-
optimism by using 100-times bootstrapping and 100-fold LGOCV method was summarized in
S5 Table in S1 File. The appearing frequency of each selected radiomics feature and optimism-
corrected AUC (bootstrap: 0.9; LGOCV: 0.898) represent an acceptable reliability of the fea-
ture and the predictive capability of constructed radiomics model.

**Table 2. The coefficient of each feature in different logistic regression models.**

| Model A[a] | |
|---|---|
| **variables** | **Coef.** |
| intercept | 6.8715 |
| wavelet-LHL_firstorder_Skewness | -3.1308 |
| wavelet-HHL_glcm_Idn | 9.1974 |
| wavelet-LLL_glszm_SizeZoneNonUniformityNormalized | -3.6223 |
| wavelet-LLH_glcm_InverseVariance | -4.8207 |
| wavelet-HLH_gldm_DependenceNonUniformityNormalized | -3.7400 |
| wavelet-HHH_glcm_MCC | 0.6721 |
| wavelet-HLH_glszm_SmallAreaEmphasis | 4.7616 |
| original_shape_Flatness | -4.8054 |
| original_shape_MajorAxisLength | -0.3870 |
| original_glszm_SizeZoneNonUniformity | 4.9277 |
| original_firstorder_10Percentile | -4.0563 |
| wavelet-HHH_glcm_Imc1 | -11.1807 |
| wavelet-HHH_glszm_GrayLevelNonUniformityNormalized | 1.7932 |
| **Model B** | |
| **variables** | **Coef.** |
| intercept | -0.5569 |
| Volume ratio of bottom-right lung lobe | 0.0700 |
| Volume ratio of bottom-left lung lobe | 0.0586 |
| **Model C** | |
| **variables** | **Coef.** |
| intercept | -2.5602 |
| Middle-right lung lobe | 0.3299 |
| Upper-left lung lobe | -0.2769 |
| Upper-right lung lobe | 0.2282 |
| Bottom-right lung lobe | 1.1397 |
| Bottom-left lung lobe | 0.8675 |
| **Model D** | |
| **variables** | **Coef.** |
| intercept | -0.1688 |
| Left_GGO | 0.0693 |
| Right_GGO | 0.0644 |
| **Model E** | |
| **variables** | **Coef.** |
| intercept | 0.2100 |
| CT lesion score (Model C score) | 1.3920 |
| Component score (Model D score) | -1.0101 |

[a]: Model A: radiomics model; Model B: lesion volume ratio model; Model C: lesion score model; Model D: lesion component ratio model; Model E: combined model of lesion score and lesion component ratio.

For CT volume-ratio based models, the Model E which combining lesion score and component ratio performed better than others, with the AUC, sensitivity, specificity and accuracy of 0.84, 0.692, 0.853, 0.756 in the training set and 0.779, 0.667, 0.826, 0.729 in the testing set. It could be observed that the scored lesion volume ratio could enhance the differentiation performance than lesion volume ratio in its absolute value. Meanwhile, by

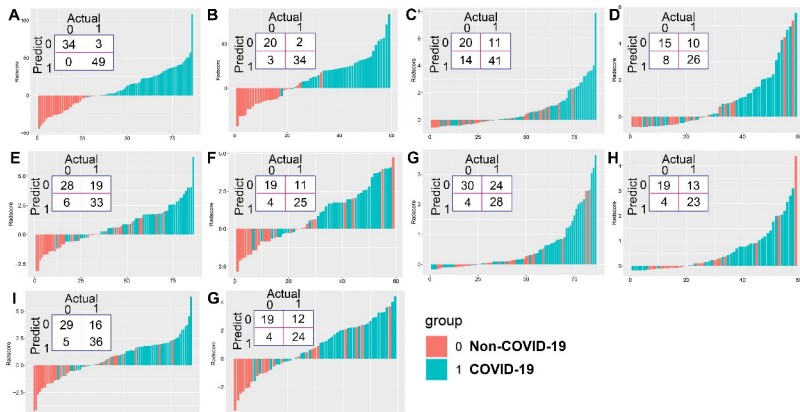

**Fig 3. The confusion matrix and the logistic model score of each model in differentiating non-Covid-19 (NC) and Covid-19 groups.** (A, B) Training and testing sets of Model A: radiomics model; (C, D) Training and testing sets of Model B: lesion volume ratio model; (E, F) Training and testing sets of Model C: lesion score model; (G, H) Training and testing sets of Model D: lesion component ratio model; (I, J) Training and testing sets of Model E: combination model using lesion score and component ratio.

further combining the ratio score and component ratio features, the model performance could be further enhanced.

After the Delong test, the statistical differences in ROC of each model were summarized in S6 Table in S1 File. It can be seen that the imaging radiomics Model A is statistically different from all other models (P <0.05). While the best volume-ratio-based Model E (combined model) only showed statistically differences compared with Model B and Model D in the training sets. And there was no significant difference between Model C (lesion score) and combined Model E.

The analysis of the calibration curve showed that the radiomics Model A (C-index: 0.994 in the training group; 0.977 in the testing group) and combined Model E(C-index: 0.840 in the

**Table 3. The performance of each classification model in the training and testing sets.**

| Model | Cut-value | Training set | | | | Testing set | | | |
|---|---|---|---|---|---|---|---|---|---|
| | | AUC | Sensitivity | Specificity | Accuracy | AUC | Sensitivity | Specificity | Accuracy |
| | | (95%CI) | (95%CI) | (95%CI) | (95%CI) | (95%CI) | (95%CI) | (95%CI) | (95%CI) |
| **Model A: radiomics model** | >0.66 | 0.994 | 0.942 | 1.0 | 0.965 | 0.977 | 0.944 | 0.870 | 0.915 |
| | | (0.984–1.0) | (0.8375–0.9862) | (0.8793–1.0) | (0.8982–0.9923) | (0.947–1) | (0.8091–0.9941) | (0.6703–0.9631) | (0.8125–0.9673) |
| **of Model B: lesion volume ratio model** | >-0.0532 | 0.739(0.635–0.844) | 0.788 | 0.588 | 0.709 | 0.738 | 0.722 | 0.652 | 0.694 |
| | | | (0.6580–0.8792) | (0.4220–0.7366) | (0.6056–0.7951) | (0.603–0.873) | (0.5586–0.8430) | (0.4478–0.8130) | (0.5680–0.7980) |
| **Model C: lesion score model** | >0.7429 | 0.82(0.733–0.908) | 0.635 | 0.823 | 0.674 | 0.772 | 0.694 | 0.826 | 0.712 |
| | | | (0.4984–0.7523) | (0.6611–0.9203) | (0.6056–0.7951) | (0.638–0.905) | (0.5303–0.8211) | (0.6226–0.9363) | (0.6211–0.8404) |
| **Model D: lesion component ratio model** | >0.3553 | 0.682(0.566–0.798) | 0.538 | 0.882 | 0.674 | 0.681 | 0.639 | 0.826 | 0.712 |
| | | | (0.4050–0.6666) | (0.7278–0.9593) | (0.5695–0.7644) | (0.535–0.828) | (0.4752–0.7758) | (0.6226–0.9363) | (0.5855–0.8123) |
| **Model E: combination model** | >0.9339 | 0.84(0.754–0.925) | 0.692 | 0.853 | 0.756 | 0.779 | 0.667 | 0.826 | 0.729 |
| | | | (0.5566–0.8015) | (0.6939–0.9403) | (0.6547–0.8350) | (0.652–0.906) | (0.5026–0.7986) | (0.6226–0.9363) | (0.6032–0.8265) |

training group; 0.779 in the testing group) had good predictive performance in the training and testing group (S8 Fig in S1 File). The decision curve analysis also indicated that the use of radiomics model A to differentiate COVID-19 from NC might achieve more net benefit (>0.85) than using the full regimen treatment or no treatment for patients in the threshold range of 0.2–0.8, which possess a higher clinical application value (Fig 4).

## The statistics of CT manifestation and signs distributions in COVID-19 and NC

In order to find the association between the CT manifestations or signs and different pneumonia types, the distribution of CT manifestations and signs were compared using chi-squared or Fisher's exact test, including lesion distribution, GGO involvement pattern, lobe predomination, lesion contour, bronchial wall thickening, air bronchogram, tree in bud sign, interlobular septal thickening, Intralobular septal_thickening, pleura effusion, and lesion attenuation.

The counts and percentage of individual CT sign in each group were summarized in S7 Table in S1 File. In the training set, the lesion distribution, GGO involvement pattern, lesion contour, bronchial wall thickening, air bronchogram and tree in bud sign showed significant differences (P < 0.05) between COVID-19 and NC patients. While in the testing group, the significant differences existed in lesion distribution, GGO involvement pattern, lobe predomination, bronchial wall thickening, tree in bud sign and intralobular septal thickening. Considering the same significant CT sign which simultaneously existed in training and testing set, it could be found from the Cramer's V value that lesion distribution (Cramers' V in training and testing: 0.792, 0.596), bronchial wall thickening(Cramers' V in training and testing: 0.591, 0.414), tree in bud sign (Cramers' V in training and testing:0.487, 0.788) and GGO involvement pattern (Cramers' V in training and testing: 0.371, 0.686) showed association with

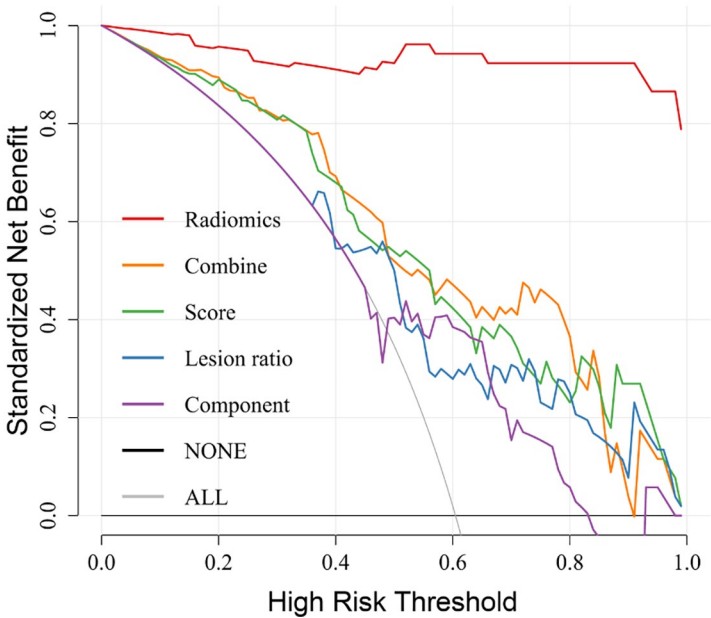

**Fig 4. The decision curves of different models.** The y-axis represents the net benefit. The y-axis represents the net benefit. Model A: red; Model B: blue; Model C: green; Model D: purple; Model E: orange. The solid grey line represents the hypothesis that all patients suffered from COVID-19 (ALL). The solid black line represents the hypothesis that no patient receives treatment (NONE). At any given threshold, the highest curve (radiomics Model A) is the optimal decision-making strategy to maximize net benefits compared to other models.

different pneumonia of COVID-19 and NC. It seems that the bronchial wall thickening and tree in bud sign happened in NC more frequently, compared with that of COVID-19 patients. In addition, from the aspect of adjusted residuals, the lesion central distribution more likely happened in NC patients while lesion peripheral distribution was more frequent in COVID-19 patients. Meanwhile, GGO involvement pattern of Patchy GGO appeared more frequently in COVID-19 patients while GGO involvement of Cluster-like GGO was more frequent in NC patients.

## Discussion

For CT signs, both influenza and COVID-19 presented GGO or GGO with consolidation. And multifocal lesions, GGO, and consolidation are the common CT findings of viral pneumonia [20–22]. Nevertheless, there are some differences in the CT manifestations between COVID-19 and influenza virus pneumonia. COVID-19 presented a high frequency of clear lesion margin and shrinking contour compared with influenza pneumonia. In the aspect of clinicopathological findings, COVID-19 histological examination showed bilateral diffuse alveolar damage with cellular fibromyxoid, desquamation of pneumocytes and hyaline membrane, and pulmonary edema with hyaline membrane formation, suggestive of early-phase ARDS formation. The clinicopathological features of influenza pneumonia findings were bronchiolitis with hemorrhage and necrotic debris, diffuse alveolar damage and alveolar hemorrhage. In addition, organizing pneumonia with prominent fibrin exudation and pulmonary edema were also observed. Irregularly fragmented hyaline membranes were seen at the alveolar orifice. The similar pathological features of COVID-19 as those in SARS and MERS also complicate the discrimination between COVID-19 and NC.

The selected subjects of this study are the patients who were difficult to diagnose after the consultation of MDT in each hospital. For these highly suspected patients, our study found that using the AUC of the ROC curve as the evaluation criterion, the radiomics model using quantitative imaging features has a higher diagnostic efficacy (AUC = 0.994 (0.984–1.0)), while the other four models based on traditional quantitative CT features were slightly lower. Additionally, among all the models based on lesion or component volume ratio, the diagnostic efficiency of the single-factor model was not as good as the combined model. These results indicated that quantitative imaging analysis could obtain more comprehensive disease imaging information, which includes first-order, second-order and transformed higher-order imaging information. While this imaging information cannot be found by traditional manual interpretation.

Radiomics has played an active role in tumor researches. Recently, it was also adopted in studies discriminating pneumonia and other lung diseases. T. Zhang et al. [12] selected four independent prediction features (2 wavelet transform first-order energy features; 1 texture feature and 1 shape feature) for differentiation between focal tissue pneumonia (FOP) and peripheral lung adenocarcinoma (LA). A good model effect was obtained: AUC (0.956), accuracy (87.6%), sensitivity (85.3%) and specificity (89.7%). For the differentiation between pneumonia and tumors, lymphocyte recruitment or tumor cell proliferation may lead to a higher degree of image discrimination, so the use of lower-order radiomics features may produce better discrimination [12, 23]. However, the identification of different types of pneumonia may require more advanced quantitative radiomics features due to the higher degree of imaging overlap. Sang Ok Park et al. [10] defined the texture of normal tissues, ground glass density shadow tissue, mesh tissue, honeycomb tissue and emphysema or solid tissue, and applied the volume proportion of each texture to achieve a model with 82% accuracy distinguishing common interstitial pneumonia (UIP) and non-specific interstitial pneumonia (NSIP). In addition

to the classification of the same type of pneumonia, CT features were also used to distinguish between different types of pneumonia. For example, Bei Wang et al. [11] used radiomics to distinguish between children's pulmonary tuberculosis and community-acquired pneumonia, hoping to help overcome the difficulty of sampling children's tuberculosis and imaging overlaps. Five radiomics features (GLCM or GLRLM texture features: 4; first-order statistical feature: 1) in the lung solid part and 6 features (shape feature: 4; GLRLM texture feature: 1; first-order statistical feature: 1) in lymph nodes were screened out, respectively. Compared with the diagnosis performance of senior radiologists (AUC: 0.791), the model combining radiomics and clinical features could well improve the diagnostic performance with AUC of 0.971. These studies indicated the active role of the imaging textures in pneumonia differentiation.

In our study, the wavelet-HHL_glcm_Idn, wavelet-LLL_glszm_SizeZoneNonUniformity-Normalized, wavelet-LLH_glcm_InverseVariance and wavelet-HHH_glcm_Imc1 features showed significant differences (P<0.05) between COVID-19 and NC in both of training and testing sets. The selected features with statistical differences mainly focused in the texture features with wavelet transform. And these features also shared relatively higher absolute value for their logistic regression coefficients. For example, the wavelet-HHH_glcm_Imc1, wavelet-HHL_glcm_Idn and wavelet-LLH_glcm_InverseVariance have regression coefficients as -13.961, 9.42 and -6.056. In addition, the 100-times bootstrapping and LGOCV also indicated that these features showed their importance or higher appearing frequency (greater than 60/100, especially 93/100 and 82/100 for wavelet-HHH_glcm_Imc1 and wavelet-LLH_glcm_InverseVariance respectively) during the multiple split of dataset into training and testing set. However, compared with texture features, the first-order statistical features' contribution was weaker. For example, for the first-order features of original_firstorder_10Percentile and wavelet.LHL_firstorder_Skewness, their appearing frequency or the weight in the logistic regression were less.

The first-order features can be used to describe some pathological features. For example, relative to normal lung parenchyma, alveolar damage and abnormal transparency caused by emphysema can be well reflected by the significantly reduced average attenuation. And the tissue of idiopathic pulmonary fibrosis can also be reflected by the decreased peak sharpness of intensity distribution curve [11]. Skewness, kurtosis, and entropy have also been used to differentiate lung malignancy [24]. However, first-order statistical features cannot describe the spatial relationship between voxel features. Therefore, in order to explore the heterogeneous distribution of different tissues spatially and their correlations, it is necessary to introduce second-order texture features or filtered conversion to highlight or emphasize texture features in certain frequency domains. We found that among the selected radiomics features, the gray level co-occurrence matrix feature extracted from wavelet-transformed image-wavelet-LLH_glcm_InverseVariance describes the local flatness or fluctuation of intensity, and its relatively lower value in COVID-19 indicated that the local inhomogeneity in the COVID-19 lesion was slightly higher than that of NC. However, from the overall texture of the lesion or the textures extracted from different wavelet-filtering channel, the lesion texture manifestation might be different. Wavelet-LLL_glszm_SizeZoneNonUniformityNormalized, wavelet-HHH_glcm_Imc1 and wavelet-HHL_glcm_Idn, respectively described intensity homogeneity among zone size volumes, mutual-information-based intensity dependence which is a measure of texture complexity and normalized inverse difference which measures intensity homogeneity. Compared with NC lesions, the relatively lower value of wavelet-LLL_glszm_SizeZoneNonUniformityNormalized, wavelet-HHH_glcm_Imc1 and higher value of wavelet-HHL_glcm_Idn, in COVID-19 patients indicated the more homogeneous texture property in the image of these wavelet transformation channels. Although it still need detailed biological or physiological explanations for these texture manifestations, they

might be potential to reflect the CT signs of COVID-19 and NC from a quantitative perspective. For example, the more frequent central distribution, Cluster-like GGO or tree in bud sign in NC patients may be related to its higher texture unevenness. Therefore, the use of quantitative features might help distinguish the signs of different pneumonia in more detail, which can obtain the higher discrimination efficiency and quantify the manually interpreted CT signs. It could be suspected that the significance of the quantitative CT features might increase with the imaging overlap between different type of pneumonia, especially when it is not enough to distinguish by the CT manifestations.

Although the model using quantitative CT imaging features presented better performance, the misjudgments still existed which resulted in the imbalance of sensitivity and specificity. It also suggested that the representative quantitative imaging features should be considered from the perspective of individualization, and extracted from standardized big data. In addition, with assistance of artificial intelligence, specific clinical work still requires clinicians to make comprehensive judgments on patients based on their own experience and other auxiliary examinations.

In this study, automatic lesion extraction and manual modification were used to ensure the consistency and repeatability to a certain extent. On such basis, quantitative parameter extraction was carried out, including the volume ratio of the lesion in the lung lobe, the volume ratio of the density component and the quantitative radiomics features. We found that compared to common quantitative parameters, such as volume ratio and density component ratio, the model established by using radiomics can distinguish NC and COVID-19 more accurately, which also means that it is necessary for CT images interpretation transform from qualitative to quantitative. Although based on WHO guidelines, the diagnosis of COVID-19 is referred to comprehensive judgments such as epidemiology, clinical manifestations, nucleic acid testing, and imaging, and the imaging alone cannot be used to diagnose. Such CT radiomics model might be potential to provide additional information indicating the risk of COVID-19 especially when the nucleic acid testing was negative and the CT manifestation was overlapped with other pneumonia. In the current study, it took about 2 min 30 s per patient for delineation of pneumonia lesion VOI, 44s per patient to extract 851 radiomics features from each pneumonia VOI and less than 10s to output the risk probability being COVID-19 based on the model (the computer processor: Intel(R) Xeon (R) E-2276M CPU@2.8GHz). Therefore, if the model was validated as reliable as possible, the clinical practice could be imagined. Once the radiologist gets the patient's CT images and input it into auto-processing platform, the risk probability for the patient could be outputted within 3 min 30s. If the imaging model is highly indicative of the risk of COVID-19, repeating the nucleic acid test multiple times is recommended. Moreover, different for image-based deep learning algorithm, the calculated and saved radiomics features could help further correlated with or explain the biological or pathological manifestations in a describable and mathematical manner. The limitations of this study are as follows. Firstly, the data of patients with COVID-19 pneumonia came from four medical centers in Sichuan Province. Some scan parameters are inconsistent, and the enrolled sample size for the current imaging study was not so large. And such small sample size was not supportive to further discriminate different infections with different microbiological evidence. Secondly, our patients were mainly mild in symptom, with very a few serious patients. Thirdly, our model is not applicable to COVID-19 patients with negative chest CT imaging. Meanwhile, the method of automatically delineating the lesions and manually modifying the boundary was used. In order to make the quantitative CT indicators more standard, the unified definition of lesion boundaries is required to make through big data and multi-center standards. In addition, although quantitative imaging features can improve the identification ability of the model,

we still need to study the pathophysiological changes represented by these selected imaging features in detail.

## Conclusion

The quantitative CT imaging features are helpful to improve the diagnosis efficiency of suspected COVID-19 patients during the epidemic. The differentiation between COVID-19 and NC could be well improved by using radiomics features, compared with traditional CT quantitative values such as lesion volume ratio or component density ratio.

## Supporting information

**S1 File. Supporting information & supporting information of methods.**
(PDF)

## Author Contributions

**Conceptualization:** Shengkun Peng.

**Data curation:** Shengkun Peng, Siyun Liu.

**Formal analysis:** Siyun Liu.

**Funding acquisition:** Bo Gong.

**Investigation:** Bo Gong, Hong Pu.

**Methodology:** Xiaobo Huang, Siyun Liu, Hong Pu, Jie Zeng.

**Project administration:** Lingai Pan.

**Software:** Yang Guo.

**Supervision:** Lingai Pan, Jianxin Huang, Hong Pu, Jie Zeng.

**Validation:** Jianxin Huang.

**Writing – original draft:** Yang Guo, Siyun Liu.

**Writing – review & editing:** Shengkun Peng, Lingai Pan, Yang Guo, Xiaobo Huang.

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
