## [Decision Letter · Decision Letter 0]

12 Apr 2021

PONE-D-20-25966

Quantitative CT imaging features for COVID-19 evaluation: the ability to differentiate COVID-19 from Non- COVID-19 (Highly Suspected) pneumonia patients during the epidemic period

PLOS ONE

Dear Dr. Zeng:

Thank you for submitting your manuscript to PLOS ONE. After careful consideration, we feel that it has merit but does not fully meet PLOS ONE’s publication criteria as it currently stands. Therefore, we invite you to submit a revised version of the manuscript that addresses the points raised during the review process.

Specific comments for the inclusion of additional information are provided by the reviewers, who both suggested major revisions.

We look forward to receiving your revised manuscript.

Kind regards,

Gayle E. Woloschak, PhD

Academic Editor

PLOS ONE

Additional Editor Comments:

Both reviewers had suggested major revisions as noted from the comments. Specific comments involve the inclusion of additional information for the reader.

Journal Requirements:

a) Did participants provide their written or verbal informed consent to participate in this study?

We note that one or more of the authors are employed by a commercial company: GE healthcare (China).

3.1. Please provide an amended Funding Statement declaring this commercial affiliation, as well as a statement regarding the Role of Funders in your study. If the funding organization did not play a role in the study design, data collection and analysis, decision to publish, or preparation of the manuscript and only provided financial support in the form of authors' salaries and/or research materials, please review your statements relating to the author contributions, and ensure you have specifically and accurately indicated the role(s) that these authors had in your study. You can update author roles in the Author Contributions section of the online submission form.

3.2. Please also provide an updated Competing Interests Statement declaring this commercial affiliation along with any other relevant declarations relating to employment, consultancy, patents, products in development, or marketed products, etc.  

Reviewers' comments:

Reviewer's Responses to Questions

**Comments to the Author**

1. Is the manuscript technically sound, and do the data support the conclusions?

Reviewer #1: Yes

Reviewer #2: Yes

2. Has the statistical analysis been performed appropriately and rigorously? 

Reviewer #1: Yes

Reviewer #2: Yes

3. Have the authors made all data underlying the findings in their manuscript fully available?

Reviewer #1: Yes

Reviewer #2: Yes

4. Is the manuscript presented in an intelligible fashion and written in standard English?

Reviewer #1: No

Reviewer #2: Yes

5. Review Comments to the Author

Reviewer #1: This manuscript describes how radiomics-based logistic regression model can differentiate COVID-19-associated pneumonia from non COVID-19 pneumonia when patients present positive CT findings. While the finding is promising, it is still questionable how reproducible the result is because the technical details do not seem to be sufficient, particularly the software package utilized. There was a very similar study recently published using the datasets acquired in a different Chinese province as well. Significance and scientific rigor will be strengthened if the paper describes how the software works, and discussion to potentially test the models described in this paper and the other paper against the data although it is not required because of potential challenges to acquire the data from other sides. Overall, there is a clear technical merit for the study presented in this manuscript which can be improved with further revision.

Specific comments are:

1. Line 162: The reference [11] does not seem to be connected with the sentence “Among the rest population, …”

2. Lines 187-188: Please correct the unit for x-ray tube voltage: 130kv to 130kVp, 120 kV to 120 kVp.

3. Line 197 (and thereafter): Please consider changing from “Imaging analysis” to “Image analysis”

4. Line 199: Is “Lung Kit” software available commercially from GE Healthcare? It is hard to find any information about it except for papers published by authors including an author from GE. If it is not available, the functionality and other important details for this software, for reproducibility, better be described.

5. Lines 198-210: The writing of radiomics processing is basically identical to similar papers, particularly one published in 2021 (Huang Y, et al. MBC Medical Imaging). Although it is understood that the same software and same processing method were used for different patient population, the only significance difference seems to be where the datasets were obtained (Yunnan vs. Sichuan). Significance of this paper is diminished by not including all available datasets, and repeating the same technique for the same kind of data – resulting in slightly different performances. It is imperative to make a good rationale over what was presented and found in the Hunag 2021 paper.

6. About the above point, it is a good discussion topic to be included to test the radiomics-based model in those already collected datasets in a different Chines province.

7. A discussion about how this radiomics-based model could be added for its utility in comparison to RT-PCR. For example, if a patient is presented with pneumonia, and positive CT findings, but initial PCR testing is negative, this model could be used to further assure the negative or positive COVID-19 diagnosis, which could potentially help select a more appropriate treatment course.

8. There are many grammatical errors throughout the manuscript. A thorough copyediting is strongly recommended for revision.

Reviewer #2: The objective of the study is to investigate the effectiveness of CT-derived image features (Radiomics) in discriminating COVID-19 from non-COVID Pneumonia (NC). The study includes 88 cases with COVID and 57 patients with NC. The radiomics features were selected sequentially based on correlation analysis, Mann-Whitney U test and the classification was performed by backward stepwise LASSO logistic regression models. The performance of radiomics feature subset were further compared with traditional CT lesion score and volumes. The classification model with radiomics features outperformed traditional CT measures with an Area Under the Curve (AUC) of 0.97 vs. 0.77 (combination model of lesion related features). Although the presented results clearly indicate the radiomics feature subset’s superior performance, there are few major concerns regarding feature selection process and model evaluation.

Major Comments:

1. The authors mention that the dataset is randomly divided into training and test set using the ratio of 3:2. Is the randomized test set consists of reasonable class distribution (stratified)?

2. It would help understand model robustness if authors can add precision-recall plots for each model.

3. It would add a great value to the manuscript if author can discuss the importance of selected features from the radiomics pool. Are the features selected were consistent across multiple splits of train and test?

4. It is not clear whether the feature selection was performed on training set only? Or was the process performed on all available instances prior to train/test division? This would affect the performance of selected features on the test set.

Minor Comments:

1. Line 114. Need to provide a citation

2. Line 121 Please mention full description of GGO as it is mentioned for the first time

6. PLOS authors have the option to publish the peer review history of their article (what does this mean?). If published, this will include your full peer review and any attached files.

Reviewer #1: No

Reviewer #2: **Yes: **Sandeep Bodduluri

---

## [Author Response · Author response to Decision Letter 0]

5 Jun 2021

Answer to Reviewer #1:

We sincerely thank the reviewer for the hard work and valuable suggestions.

Reviewer #1: This manuscript describes how radiomics-based logistic regression model can differentiate COVID-19-associated pneumonia from non COVID-19 pneumonia when patients present positive CT findings. While the finding is promising, it is still questionable how reproducible the result is because the technical details do not seem to be sufficient, particularly the software package utilized. There was a very similar study recently published using the datasets acquired in a different Chinese province as well. Significance and scientific rigor will be strengthened if the paper describes how the software works, and discussion to potentially test the models described in this paper and the other paper against the data although it is not required because of potential challenges to acquire the data from other sides. Overall, there is a clear technical merit for the study presented in this manuscript which can be improved with further revision.

Specific comments are:

1. Line 162: The reference [11] does not seem to be connected with the sentence “Among the rest population, …” 

Response:

We thank the reviewer for the kind reminder. The reference [11] was misplaced here so we have removed the reference [11] in the revised version.

2. Lines 187-188: Please correct the unit for x-ray tube voltage: 130kv to 130kVp, 120 kV to 120 kVp. 

Response:

We thank the reviewer for this kind reminder. We have corrected the unit of X-ray tube voltage from kV into kVp.

3. Line 197 (and thereafter): Please consider changing from “Imaging analysis” to “Image analysis” 

Response:

Thank you very much for this kind suggestion. We have changed the description from “Imaging analysis” into “Image analysis”.

4. Line 199: Is “Lung Kit” software available commercially from GE Healthcare? It is hard to find any information about it except for papers published by authors including an author from GE. If it is not available, the functionality and other important details for this software, for reproducibility, better be described.

Response:

We thank the reviewer for this valuable question. “Lung Kit” software (GE Healthcare) is one commercialized software only for scientific research. The software contains analysis module for COVID-19 in which the four-step data processing flow includes:

 (1) user-defined image preprocessing: such as resampling, intensity discretization, image filtering or denoising and so on. In the current research, all the images were firstly resampled into isotropic voxel size of 1 mm*1 mm*1 mm using trilinear interpolation. After interpolation into isotropic voxel, the image’s intensity values were rounded to the nearest integer HU value. The low-pass Gaussian filter with σ = 0.5 was then conducted to increase the reproducibility of the radiomics features. 

(2) the automatic segmentation of lung lobes and pneumonia lesion: the five lung lobes were segmented based on deep learning algorithm Dense V-networks [Gibson, E., Giganti, F., Hu, Y., Bonmati, E., Bandula, S., & Gurusamy, K. (2018). Automatic Multi-organ Segmentation on Abdominal CT with Dense V-networks. 0062(c), 1–12. https://doi.org/10.1109/TMI.2018.2806309 ], including 2 lobes in left lung (upper, bottom) and 3 lobes in right lung (upper, middle, bottom). Based on the lung lobes segmentation, the pneumonia lesions were detected and volume of interest (VOI) was segmented as a whole. The machine-human collaboration was applied to guarantee correct segmentation as described in the main text and we also carried it here for reviewer’s convenience:

“The margin of the VOI was checked and manually adjusted by an experienced thoracic radiologist (SK. P, a radiology attending doctor with 7 years’ experience in interpreting chest CT images) and the obviously swollen blood vessels involved in the lesion were excluded, if necessary. All the automatically segmented or manually adjusted VOIs were checked by a senior radiologist (H. P [a thoracic radiologist with 28 years’ experience]) to reach consensus. The distributed lesions were considered as a whole VOI in the following analysis steps. “

(3) quantization of pneumonia lesion: the volume of each segmented lung lobe and pneumonia lesion were firstly calculated. The lesion volume ratio and lesion component analysis were conducted as described in the main text and we also carried it here for reviewer’s convenience.

“For lesion volume ratio analysis, the lesion volume ratio in five lung lobes was respectively calculated automatically by LK2.2 after lesion VOI was delineated. The lesion ratio in each lung lobe was scored from 0 to 5 which was defined according to the volume ratio involved: 0, no lesion; 1, ≤5%; 2, 6%-25%; 3, 26%-49%; 4, 50%-75%; 5, >75%[ F P, T Y, P S, S G, B L, L L, et al. Time Course of Lung Changes at Chest CT during Recovery from Coronavirus Disease 2019 (COVID-19). Radiology. 2020;295(3):715-21.].

For lesion component analysis, the part solid component was defined as the lesion with CT value <-200 HU while the parts with CT value >-200 HU were defined as solid component. And the volume ratio of each kind of lesion component in the left and right lung was derived automatically.”

(4) radiomics feature extraction based on the lesion VOIs: 

The “Lung Kit” software carried open source of Python package Pyradiomics v2.2 (https://pyradiomics.readthedocs.io/en/latest/index.html) and provide user interface to set the parameters for feature extraction. In the current study, following the CT value discretized with binWidth = 25HU, a total of 851 radiomics features were extracted from segmented pneumonia VOIs [Larue RTHM, van Timmeren JE, de Jong EEC, Feliciani G, Leijenaar RTH, Schreurs WMJ, Sosef MN, Raat FHPJ, van der Zande FHR, Das M, van Elmpt W, Lambin P. Influence of gray level discretization on radiomic feature stability for different CT scanners, tube currents and slice thicknesses: a comprehensive phantom study. Acta Oncol. 2017 Nov;56(11):1544-1553. doi: 10.1080/0284186X.2017.1351624. PMID: 28885084 .]. 

In brief, a total of 851 extracted radiomics features were categorized into three groups: first-order features, textural features, and transformed features. There were 32 first-order features consisting of 18 intensity statistical features and 14 morphological features. Among 75 textural features, there were 24 Gray Level Co‐occurrence Matrix (GLCM), 16 Gray Level Run Length Matrix (GLRLM), 16 Gray Level Size Zone Matrix (GLSZM), 14 Gray Level Dependence Matrix (GLDM) and 5 Neighboring Gray Tone Difference Matrix (NGTDM) features. For the transformed images, first-order wavelet transform decomposed the ROI into 8 sub-VOIs by using either a high- or low-pass filter in three dimensional directions, including high–high–high, high–high–low, high–low–low, high–low–high, low–high–low, low–high–high, low–low–high, and low–low–low. The same set of texture features (18 intensity statistical features, 24 GLCM features, 16 GLRLM, 16 GLSZM, 14 GLDM, and 5 NGTDM) were calculated based on these wavelet-transformed images and 744 wavelet features were obtained finally. 

And Lung Kit software has been applied in texture-based analysis of severity or prognosis and differentiation from influenza pneumonia in some published manuscript as well. [Li L, Wang L, Zeng F, Peng G, Ke Z, Liu H, Zha Y. Development and multicenter validation of a CT-based radiomics signature for predicting severe COVID-19 pneumonia. Eur Radiol. 2021 Mar 30:1–12. doi: 10.1007/s00330-021-07727-x. PMID: 33786655; PMCID: PMC8009273.][Huang Y, Zhang Z, Liu S, Li X, Yang Y, Ma J, Li Z, Zhou J, Jiang Y, He B. CT-based radiomics combined with signs: a valuable tool to help radiologist discriminate COVID-19 and influenza pneumonia. BMC Med Imaging. 2021 Feb 17;21(1):31. doi: 10.1186/s12880-021-00564-w. PMID: 33596844; PMCID: PMC7887546.][Ke Z, Li L, Wang L, Liu H, Lu X, Zeng F, Zha Y. Radiomics analysis enables fatal outcome prediction for hospitalized patients with coronavirus disease 2019 (COVID-19). Acta Radiol. 2021 Feb 18:284185121994695. doi: 10.1177/0284185121994695. Epub ahead of print. PMID: 33601893.][Xiao F, Sun R, Sun W, Xu D, Lan L, Li H, Liu H, Xu H. Radiomics analysis of chest CT to predict the overall survival for the severe patients of COVID-19 pneumonia. Phys Med Biol. 2021 Apr 12. doi: 10.1088/1361-6560/abf717. Epub ahead of print. PMID: 33845467.][Wei W, Hu XW, Cheng Q, Zhao YM, Ge YQ. Identification of common and severe COVID-19: the value of CT texture analysis and correlation with clinical characteristics. Eur Radiol. 2020 Dec;30(12):6788-6796. doi: 10.1007/s00330-020-07012-3. Epub 2020 Jul 1. PMID: 32613287; PMCID: PMC7327490.] 

5. Lines 198-210: The writing of radiomics processing is basically identical to similar papers, particularly one published in 2021 (Huang Y, et al. MBC Medical Imaging). Although it is understood that the same software and same processing method were used for different patient population, the only significance difference seems to be where the datasets were obtained (Yunnan vs. Sichuan). Significance of this paper is diminished by not including all available datasets, and repeating the same technique for the same kind of data – resulting in slightly different performances. It is imperative to make a good rationale over what was presented and found in the Hunag 2021 paper.

Response:

Thank you very much for this valuable suggestion to help us strengthen the manuscript. Firstly, we feel sorry about making the reviewer confusing about the study object of these two studies. And we would like to describe the difference from the Huang’s study and significance of the current manuscript from two aspects.

(1) The patient population and study object of the current research

The etiology was different for the patient population in the current research. The pulmonary appearance of COVID-19 is very similar to that of influenza or some other mixed infection. Our control group patients are more representative of the majority of the population and are more universal. Among the patients in our control group, there are viral infections, bacterial infections, and mixed infections, which are more diverse. In addition, all the patients were collected during the COVID-19 epidemic. This kind of differentiation research might better reflect its value during the COVID-19 epidemic. Because in fact, when we encountered community-acquired pneumonia at the very beginning, it might be not possible to immediately diagnose whether the patient is infected with the virus. Besides, the same pathogenic infection may show different clinical manifestations and imaging characteristics in different regions. And the patients with independent pathogenic infection were relatively rare during the same epidemic to proceed the differentiation . So it is meaningful for us to compare COVID-19 patients and patients with other pathogenic infections during the COVID-19 epidemic. And the object of the current research was to comprehensively consider different kinds of quantitative CT imaging features and to evaluate their differentiation performance for the COVID-19 from other pathogenic infections during the COVID-19 epidemic.

(2) The image-based features involved in the current research

In the current study, one of the main idea is to evaluate the performance of quantitative CT-derived features in COVID-19 differentiation. The image-based features in the current study included radiomics features and quantized CT-manifestations. Therefore, we tried to construct five CT-value-derived models: Model A was by CT radiomics features; Model B by lesion volume ratio in five lung lobes; Model C by lesion ratio score in each lung lobe derived from leveled lesion volume ratio; Model D was constructed by volume ratio of lesion part-solid and solid component in left and right lungs; Model E was combined model using lesion score and component ratio. The construction and comparison for these different models were from the consideration that whether we could find out some simple quantitative CT-derived parameters instead of relatively complex calculation to fulfill the COVID-19 differentiation. Because the simpler the method, the easier for clinical practice. Therefore, the consideration of the image-derived features was different from the other study which involved radiomics features and CT signs or their combination.

We sincerely thank the reviewer again for this kind suggestion.

6. About the above point, it is a good discussion topic to be included to test the radiomics-based model in those already collected datasets in a different Chines province.

Response:

We deeply agree with the reviewer’s suggestion that an external validation of the model performance was necessary if the patient population or the enrollment criteria was the same. However, we felt very sorry that the COVID-19 data sharing between different province is limited by the policy and the data involved in the current research is the most available data that our hospital could collect around Sichuan province. But we would keep it in mind and try to externally validate the model if we have obtained the external data for other province or we would also consider to participate in multicenter-study if there is related project proposed in our country.

7. A discussion about how this radiomics-based model could be added for its utility in comparison to RT-PCR. For example, if a patient is presented with pneumonia, and positive CT findings, but initial PCR testing is negative, this model could be used to further assure the negative or positive COVID-19 diagnosis, which could potentially help select a more appropriate treatment course.

Response:

Thank you very much for this valuable suggestion on the clinical practice of the constructed model. We have added a discussion about the clinical practice potential for the radiomics model in the revised manuscript, and we also described it here for reviewer’s convenience.

Revised manuscript on LINE 508: 

Although based on WHO guidelines, the diagnosis of COVID-19 is referred to comprehensive judgments such as epidemiology, clinical manifestations, nucleic acid testing, and imaging, and the imaging alone cannot be used to diagnose. Such CT radiomics model might be potential to provide additional information indicating the risk of COVID-19 especially when the nucleic acid testing was negative and the CT manifestation was overlapped with other pneumonia. In the current study, it took about 2 min 30 s per patient for delineation of pneumonia lesion VOI, 44s per patient to extract 851 radiomics features from each pneumonia VOI and less than 10s to output the risk probability being COVID-19 based on the model (the computer processor: Intel(R) Xeon (R) E-2276M CPU@2.8GHz). Therefore, if the model was validated as reliable as possible, the clinical practice could be imagined. Once the radiologist gets the patient’s CT images and input it into auto-processing platform, the risk probability for the patient could be outputted within 3 min 30s. If the imaging model is highly indicative of the risk of COVID-19, repeating the nucleic acid test multiple times is recommended. Moreover, different for image-based deep learning algorithm, the calculated and saved radiomics features could help further correlated with or explain the biological or pathological manifestations in a describeable and mathematical manner. 

8. There are many grammatical errors throughout the manuscript. A thorough copyediting is strongly recommended for revision.

Response:

Thank you very much for kind reminder. We have carefully checked and revised the manuscript thoroughly.

Answer to Reviewer #2:

We sincerely thank the reviewer for the hard work and valuable suggestions.

Reviewer #2: The objective of the study is to investigate the effectiveness of CT-derived image features (Radiomics) in discriminating COVID-19 from non-COVID Pneumonia (NC). The study includes 88 cases with COVID and 57 patients with NC. The radiomics features were selected sequentially based on correlation analysis, Mann-Whitney U test and the classification was performed by backward stepwise LASSO logistic regression models. The performance of radiomics feature subset were further compared with traditional CT lesion score and volumes. The classification model with radiomics features outperformed traditional CT measures with an Area Under the Curve (AUC) of 0.97 vs. 0.77 (combination model of lesion related features). Although the presented results clearly indicate the radiomics feature subset’s superior performance, there are few major concerns regarding feature selection process and model evaluation.

Major Comments:

1. The authors mention that the dataset is randomly divided into training and test set using the ratio of 3:2. Is the randomized test set consists of reasonable class distribution (stratified)?

Response:

Thank you very much for this kind reminder.

Yes, the data was divided into a training and test set at a ratio of 3:2 using a stratified random sampling method. In the training set, there were 52 COVID-19 patients and 34 non-COVID-19 patients. In the test set, there were 36 COVID-19 patients and 23 non-COVID-19 patients. 

The statistics difference of CT manifestations and signs features between training and testing set was summarized in Table C1. There was no significant bias for the CT manifestations and signs between training and test set.

Table C1 The statistics difference of CT manifestations and signs features between training and testing set.

We have also revised the manuscript (part “Data preprocessing, model establishment and evaluation”) on LINE 220 as follows: 

“The dataset (n = 145; COVID-19 = 88, NC = 57) was randomly stratified into training (n = 86) and testing (n = 59) set using the ratio of 3:2.”

2. It would help understand model robustness if authors can add precision-recall plots for each model.

Response:

We thank the reviewer for this valuable suggestion to help us strengthen the manuscript.

Following the reviewer’s instruction, we have added the precision-recall plots for 5 models constructed in the manuscript, including (1) Model A: radiomics model; (2) Model B: lesion volume ratio model; (3) Model C: lesion score model; (4) Model D: lesion component ratio model; (5) Model E: combined model using lesion score and component ratio simultaneously. And the area under the PR curve for each model were summarized in the inset. As shown in Figure S7 , all the model performance shared the same tendency as the ROC analysis. The Radiomics model (Model A) performed better than other image-derived models both in the training and test group, which was followed by combined model (Model E). It indicated the robustness of the radiomics model.

Figure S7 The Precision-Recall (PR) curve of each model in the training (A) and testing (B) sets. The area under the PR curve for each model was labeled in the legend. 

3. It would add a great value to the manuscript if author can discuss the importance of selected features from the radiomics pool. Are the features selected were consistent across multiple splits of train and test?

Response:

We thank the reviewer again for this valuable suggestion and help us check out some errors and strengthen the manuscript. 

We carefully checked the radiomics features reliability measurement in the original manuscript and found out that the description for the method was wrong which should be corrected from “100-fold leave-group-out cross-validation (LGOCV)” into “100-times bootstrapping method”. In addition, we found out that some parts of discussion about the radiomics features should be corrected or supplemented.

Therefore, besides the 100-times bootstrapping used in the original manuscript, in the revised version, we also supplemented the method of 100-fold leave-group-out cross-validation (LGOCV) with the proportion of data in the sub–training sets (60%), to measure the reliability of the radiomics features involved in the final model and the model overoptimism. The reliability of the radiomics features was measured by the appearing frequency during multiple splits of training and test sets. While the model overoptimism was verified by the mean AUC in the multiple-split training and test sets. We have corrected, revised or supplemented the corresponded context in the method, result and discussion part for the revised manuscript. And we also summarized here for reviewer’s convenience.

(1) The method for the 100-times bootstrapping and LGOCV was described as follows.

(1-1) 100-times bootstrapping method

Step 1: Apparent performance. Taking the model’s AUC performance of predicting model developed in the original whole training dataset as the apparent performance.

Step2: BOOTSTRAP sample splitting. The whole original training dataset was repeatedly split into bootstrapped training and test sets. The bootstrapped training set with the same sample size as original training dataset was constructed by sampling with replacement from the original sample. The out-of-bag samples constructed the test set in each bootstrap.

Step3: BOOTSTRAP model establishment. Starting features were features used to establish radiomics model based on the original training dataset. And these features were further selected by backward stepwise logistic regression method with minimum AIC (the same as the modeling method in the manuscript) in each bootstrapped training set. The logistic regression model established from the bootstrapped training set was respectively tested in the “out-of-bag” test set. The AUC were obtained for bootstrapped training and test set in each bootstrap loop.

Step4: Model optimism among bootstrap. Calculate the model optimism as the difference between the AUC of bootstrapped training set and the test set.

Step5: The optimism-corrected performance. Repeating Step 2 to Step 4 for 100 times. The appearing frequency of each feature and the average optimism were calculated and recorded. Subtract the value from the apparent performance in step 1 and obtain an optimism-corrected performance.

(1-2) 100-fold leave-group-out cross-validation (LGOCV) method

Step 1: Apparent performance. Taking the model’s AUC performance of predicting model developed in the original whole training dataset as the apparent performance.

Step2: Multiple splitting of training and test sets. The whole dataset was repeatedly split into training and test sets. A proportion of data in the sub–training sets was set as 60% which was the same as the randomized stratification ratio of 3:2 in the original whole dataset. The rest 40% data were taken as test set in each loop. Such group splitting was repeated 100 times.

Step3: LGOCV model establishment. Starting features were features used to establish radiomics model based on the original training dataset. And these features were further selected by backward stepwise logistic regression method with minimum AIC (the same as the modeling method in the manuscript) in each LGOCV training set. The logistic regression model established from the LGOCV training set was respectively tested in the LGOCV test set. The AUC were obtained for LGOCV training and test set in each loop.

Step4: Model optimism among bootstrap. Calculate the model optimism as the difference between the AUC of LGOCV training set and test set.

Step5: The optimism-corrected performance. Repeating Step 2 to Step 4 for 100 times. The appearing frequency of each feature and the average optimism were calculated and recorded. Subtract the value from the apparent performance in step 1 and obtain an optimism-corrected performance.

(2) The results for the 100-times bootstrapping and LGOCV were summarized as follows.

The 100-times bootstrapping result showed that all of the 10 radiomics features except for wavelet.LHL_firstorder_Skewness (47 times in 100 folds) appeared more than 50 times during 100-times bootstrapping, which indicated the reliability of the selected features. As shown in Table S3.

Table S3 The appearing frequency of the selected radiomics features during 100-times bootstrapping.

The LGOCV result showed a similar appearing frequency of features with bootstrapping method, in which all of the 10 radiomics features except for wavelet.LHL_firstorder_Skewness (43 times in 100 folds) appeared more than 50 times during 100-times group splitting. As shown in Table S4.

Table S4 The appearing frequency of the selected radiomics features during 100-fold leave-group-out cross-validation (LGOCV).

The estimation of model overoptimism by using 100-times bootstrapping and 100-fold leave-group-out cross-validation (LGOCV) method was summarized in Table S5. The appearing frequency of each selected radiomics feature and optimism-corrected AUC (bootstrap: 0.9; LGOCV: 0.898) represent an acceptable reliability of the feature and the predictive capability of constructed radiomics model. 

Table S5 Estimation of the area under the ROC curve and the model overoptimism for the radiomics model using 100-times bootstrapping and 100-fold leave-group-out cross-validation (LGOCV) method.

(3) The revised discussion about the selected radiomics features.

Revised manuscript on LINE 447 (the italic and bold font indicates the revised part):

In our study, the wavelet-HHL_glcm_Idn, wavelet-LLL_glszm_SizeZoneNonUniformityNormalized, wavelet-LLH_glcm_InverseVariance and wavelet-HHH_glcm_Imc1 features showed significant differences (P<0.05) between COVID-19 and NC in both of training and testing sets. The selected features with statistical differences mainly focused in the texture features with wavelet transform. And these features also shared relatively higher absolute value for their logistic regression coefficients. For example, the wavelet-HHH_glcm_Imc1, wavelet-HHL_glcm_Idn and wavelet-LLH_glcm_InverseVariance have regression coefficients as -13.961, 9.42 and -6.056 respectively. In addition, the 100-times bootstrapping and LGOCV also indicated that these features showed their importance or higher appearing frequency (greater than 60/100, especially 93/100 and 82/100 for wavelet-HHH_glcm_Imc1 and wavelet-LLH_glcm_InverseVariance respectively) during the multiple split of dataset into training and testing set. However, compared with texture features, the first-order statistical features’ contribution was weaker. For example, for the first-order features of original_firstorder_10Percentile and wavelet.LHL_firstorder_Skewness, their appearing frequency or the weight in the logistic regression were less.

The first-order features can be used to describe some pathological features. For example, relative to normal lung parenchyma, alveolar damage and abnormal transparency caused by emphysema can be well reflected by the significantly reduced average attenuation. And the tissue of idiopathic pulmonary fibrosis can also be reflected by the decreased peak sharpness of intensity distribution curve [Zhang T, Yuan M, Zhong Y, Zhang YD, Li H, Wu JF, et al. Differentiation of focal organising pneumonia and peripheral adenocarcinoma in solid lung lesions using thin-section CT-based radiomics. Clin Radiol. 2019;74(1):78.e23-78.e30.]. Skewness, kurtosis, and entropy have also been used to differentiate lung malignancy [Scrivener M, de Jong EEC, van Timmeren JE, Pieters T, Ghaye B, Geets X. Radiomics applied to lung cancer: a review. Translational Cancer Research. 2016;5(4):398-409.]. However, first-order statistical features cannot describe the spatial relationship between voxel features. Therefore, in order to explore the heterogeneous distribution of different tissues spatially and their correlations, it is necessary to introduce second-order texture features or filtered conversion to highlight or emphasize texture features in certain frequency domains. We found that among the selected radiomics features, the gray level co-occurrence matrix feature extracted from wavelet-transformed image- wavelet-LLH_glcm_InverseVariance describes the local flatness or fluctuation of intensity, and its relatively lower value in COVID-19 indicated that the local inhomogeneity in the COVID-19 lesion was slightly higher than that of NC. However, from the overall texture of the lesion or the textures extracted from different wavelet-filtering channel, the lesion texture manifestation might be different. Wavelet-LLL_glszm_SizeZoneNonUniformityNormalized, wavelet-HHH_glcm_Imc1 and wavelet-HHL_glcm_Idn, respectively described intensity homogeneity among zone size volumes, mutual-information-based intensity dependence which is a measure of texture complexity and normalized inverse difference which measures intensity homogeneity. Compared with NC lesions, the relatively lower value of wavelet-LLL_glszm_SizeZoneNonUniformityNormalized, wavelet-HHH_glcm_Imc1 and higher value of wavelet-HHL_glcm_Idn, in COVID-19 patients indicated the more homogeneous texture property in the image of these wavelet transformation channels. Although it still need detailed biological or physiological explanations for these texture manifestations, they might be potential to reflect the CT signs of COVID-19 and NC from a quantitative perspective. For example, the more frequent central distribution, Cluster-like GGO or tree in bud sign in NC patients may be related to its higher texture unevenness. Therefore, the use of quantitative features might help distinguish the signs of different pneumonia in more detail, which can obtain the higher discrimination efficiency and quantify the manually interpreted CT signs. It could be suspected that the significance of the quantitative CT features might increase with the imaging overlap between different type of pneumonia, especially when it is not enough to distinguish by the CT manifestations.

4. It is not clear whether the feature selection was performed on training set only? Or was the process performed on all available instances prior to train/test division? This would affect the performance of selected features on the test set.

Response:

Thank you very much for helping us make the method much clearer for the readers.

For five models constructed in the current study, each group of features were selected sequentially only in the training dataset because the distribution of test dataset was taken as unknown status. The finally selected feature names, logistic regression coefficient and ROC cut-off value determined based on the training set were copied to the test set for model’s internal validation. In addition, the radiomics features were conducted with median values replacement for missing values and Z-score normalization prior to the feature selection procedure.

We have revised the feature selection method in the manuscript (part “Data preprocessing, model establishment and evaluation”) and also copied it here for reviewer’s convenience.

Revised manuscript on LINE 219: 

Method Part: Data preprocessing, model establishment and evaluation

“The radiomics feature data was firstly preprocessed by replacing missing values with median values and z-score normalization. The dataset (n = 145; COVID-19 = 88, NC = 57) was randomly stratified into training (n = 86) and testing (n = 59) set using the ratio of 3:2.

Following the CT image analysis and data preprocessing, five logistic regression models were constructed for differentiating COVID-19 from NC. Model A was constructed by radiomics features; Model B was constructed by lesion volume ratio in five lung lobes. Model C was constructed by lesion ratio score in each lung lobe derived from lesion volume ratio. Model D was constructed by volume ratio of lesion part-solid and solid component in left and right lungs. Model E was combined model using lesion score and component ratio. The logistic regression method was used for every model construction. For every model, the features were selected sequentially only in the training dataset because the distribution of test dataset was taken as unknown status. 

For Model A, the radiomics features for model construction were selected by the procedure as follows. The redundant features were firstly reduced by correlation analysis at a cut-value of 0.6. Then Mann-Whitney U test was used to eliminate features without statistical difference in COVID-19 and NC groups. The significant level was 0.05. Next, the least absolute shrinkage and selection operator (LASSO) logistic regression method with 10-fold cross validation was used for further feature selection and regularization to improve the model accuracy and avoid overfitting. The remaining features with non-zero coefficients were kept and involved into multi-variate backward stepwise logistic regression with minimum AIC (Akaike Information Criterion) method to establish the model. Besides, to verify the reliability of selected radiomics features in the logistic regression model, 100-fold leave-group-out cross-validation (LGOCV) was performed. 

From Model B to Model D, the features of lesion ratios or scores were sequentially selected by Mann-Whitney U test and univariate logistic analysis. The features with statistically P<0.05 were used for respective model construction. And for combined Model E, the selected significant features from lesion score and component ratio were involved simultaneously.

The five logistic regression models were finally constructed based on the training set. And models’ classification performances were evaluated by receiver operating characteristic (ROC) analysis. The area under the curve (AUC), accuracy, sensitivity and specificity were derived. To internally validate the models’ performances in the test set, the retained feature names, logistic regression coefficients and the ROC cut-off value (when Youden index reached the maximum) obtained from the training set were applied in the test set. Then the corresponding predictive performance in the test data could be derived. In addition, the calibration curves and decision curve analysis (DCA) curves were calculated to assess the models’ prediction performance and clinical benefit.”

---

## [Decision Letter · Decision Letter 1]

16 Jun 2021

PONE-D-20-25966R1

Quantitative CT imaging features for COVID-19 evaluation: the ability to differentiate COVID-19 from Non- COVID-19 (Highly Suspected) pneumonia patients during the epidemic period

PLOS ONE

Dear Dr. Zheng:

Thank you for submitting your manuscript to PLOS ONE. After careful consideration, we feel that it has merit but does not fully meet PLOS ONE’s publication criteria as it currently stands. Therefore, we invite you to submit a revised version of the manuscript that addresses the points raised during the review process.

Please address the concern about the inclusion of more information about the lung kit in a revision.

We look forward to receiving your revised manuscript.

Kind regards,

Gayle E. Woloschak, PhD

Academic Editor

PLOS ONE

Journal Requirements:

Thank you for your ethics statement, "Ethics approval and consent to participate: This retrospective study was approved by the institutional Ethics Committee of Sichuan Academy of Medical Sciences & Sichuan

Provincial People's Hospital and obtained the informed consent of the patients.Please amend your current ethics statement to address the following concerns:

In the ethics statement in the manuscript and in the online submission form, please provide additional information about the patient records used in your retrospective study, including: 

Please amend your current ethics statement to address the following concerns:

a) Did participants provide their written or verbal informed consent to participate in this study?

Additional Editor Comments (if provided):

The reviewers have asked for additional information to be added about the lung kit in order to enhance reproducibility of the work.

Reviewers' comments:

Reviewer's Responses to Questions

**Comments to the Author**

1. If the authors have adequately addressed your comments raised in a previous round of review and you feel that this manuscript is now acceptable for publication, you may indicate that here to bypass the “Comments to the Author” section, enter your conflict of interest statement in the “Confidential to Editor” section, and submit your "Accept" recommendation.

Reviewer #1: (No Response)

Reviewer #2: All comments have been addressed

2. Is the manuscript technically sound, and do the data support the conclusions?

Reviewer #1: Yes

Reviewer #2: Yes

3. Has the statistical analysis been performed appropriately and rigorously? 

Reviewer #1: Yes

Reviewer #2: Yes

4. Have the authors made all data underlying the findings in their manuscript fully available?

Reviewer #1: Yes

Reviewer #2: Yes

5. Is the manuscript presented in an intelligible fashion and written in standard English?

Reviewer #1: Yes

Reviewer #2: Yes

6. Review Comments to the Author

Reviewer #1: The response to my critiques for the previous submission looks adequate and appropriate. However, the details of “Lung Kit” software, in the response letter, better be in either the manuscript’s main text or at least in the supplementary material. It is still not clear how others could reproduce the results without knowing that it can be obtained for research.

Reviewer #2: The authors satisfactorily addressed all the comments raised in the initial review and made significant changes to the manuscript accordingly.

7. PLOS authors have the option to publish the peer review history of their article (what does this mean?). If published, this will include your full peer review and any attached files.

Reviewer #1: No

Reviewer #2: **Yes: **SANDEEP BODDULURI

---

## [Author Response · Author response to Decision Letter 1]

11 Jul 2021

Dear editor,

We are very pleased to learn again from your letter about revision for our manuscript (PONE-D-20-25966R2). Thank you very much for providing us the further suggestion and the opportunity for us to resubmit our revised manuscript. We have revised the manuscript according to the comments from the reviewers. Our answers to the comments are described as follows, with revision tracks in the revised manuscript and supplementary material.

We hope that the revised manuscript could be considered acceptable for publication. We are looking forward to hearing from you.

1.) "Please amend your current ethics statement to address the following concerns:

This retrospective study was approved by the institutional Ethics Committee of Sichuan Academy of Medical Sciences & Sichuan Provincial People's Hospital and obtained the informed consent of the patients. （In the Cover Letter）

2.) manuscript format：We have modified it.

We sincerely thank the reviewer again for the kind patience and valuable suggestion during the peer-review process.

---

## [Decision Letter · Decision Letter 2]

3 Aug 2021

Quantitative CT imaging features for COVID-19 evaluation: the ability to differentiate COVID-19 from Non- COVID-19 (Highly Suspected) pneumonia patients during the epidemic period

PONE-D-20-25966R2

Dear Dr. Zheng,

We’re pleased to inform you that your manuscript has been judged scientifically suitable for publication and will be formally accepted for publication once it meets all outstanding technical requirements.

Kind regards,

Gayle E. Woloschak, PhD

Section Editor

PLOS ONE

Additional Editor Comments (optional):

Reviewers' comments:

Reviewer's Responses to Questions

**Comments to the Author**

1. If the authors have adequately addressed your comments raised in a previous round of review and you feel that this manuscript is now acceptable for publication, you may indicate that here to bypass the “Comments to the Author” section, enter your conflict of interest statement in the “Confidential to Editor” section, and submit your "Accept" recommendation.

Reviewer #1: All comments have been addressed

Reviewer #2: All comments have been addressed

2. Is the manuscript technically sound, and do the data support the conclusions?

Reviewer #1: Yes

Reviewer #2: Yes

3. Has the statistical analysis been performed appropriately and rigorously? 

Reviewer #1: Yes

Reviewer #2: Yes

4. Have the authors made all data underlying the findings in their manuscript fully available?

Reviewer #1: Yes

Reviewer #2: Yes

5. Is the manuscript presented in an intelligible fashion and written in standard English?

Reviewer #1: Yes

Reviewer #2: Yes

6. Review Comments to the Author

Reviewer #1: All comments have been addressed appropriately. The manuscript is of good quality, and no further revision seems necessary.

Reviewer #2: The authors made significant changes to the manuscript and provided with additional statistical analysis to ensure the robustness of the prediction performance, as requested in the initial review.

7. PLOS authors have the option to publish the peer review history of their article (what does this mean?). If published, this will include your full peer review and any attached files.

Reviewer #1: No

Reviewer #2: **Yes: **Sandeep Bodduluri

---

## [Editor Report · Acceptance letter]

5 Jan 2022

PONE-D-20-25966R2 

Quantitative CT Imaging Features for COVID-19 Evaluation: the Ability to Differentiate COVID-19 from Non- COVID-19 (Highly Suspected) Pneumonia Patients during the Epidemic Period 

Dear Dr. Zeng:

I'm pleased to inform you that your manuscript has been deemed suitable for publication in PLOS ONE. Congratulations! Your manuscript is now with our production department. 

Kind regards, 

on behalf of

Dr. Gayle E. Woloschak 

Section Editor

PLOS ONE